# The Effects of Climate Change on the Tagus–Segura Transfer: Diagnosis of the Water Balance in the Vega Baja del Segura (Alicante, Spain)

**Antonio Oliva Cañizares** * , **Jorge Olcina Cantos** and **Carlos J. Baños Castiñeira**

University of Alicante, 03690 San Vicente del Raspeig, Spain; jorge.olcina@ua.es (J.O.C.);
carlos.banos@ua.es (C.J.B.C.)
* Correspondence: antoniogeografia1@gmail.com

**Abstract:** Climate change is one of the most important problems facing society in the 21st century. Despite the uncertainty about the behaviour of rainfall due to climate change, what is clear is that average rainfall has been reduced in the inland areas and headwaters of Spain's river basins. The Tagus basin is one of the most affected, with implications for the Jucar and Segura basins. The working hypothesis is to corroborate with the data collected on the effects of climate change on the TTS. To this end, the following methodology has been applied: (a) analysis in the headwaters of the Tagus, using data on precipitation, surface runoff and reservoir water; (b) analysis of the resources of the Segura basin (supply and demand), based on the basin organisation's own data; (c) construction of a water balance adjusted to the Bajo Segura district (Alicante), a user of the water transferred for agricultural use. Likewise, the data provided by the basin organisation have made it possible to corroborate the data on consumption and allocation of the corresponding volumes of water. The results obtained make it possible to put forward a novel proposal in the scientific field related to hydrological planning based on the principles of sustainability.

**Keywords:** climate change; Tagus–Segura transfer; sustainable hydrological planning; Bajo Segura

## 1. Introduction

The sixth report of the Intergovernmental Panel on Climate Change (IPCC) (AR6) published in 2021 identifies the Mediterranean region (MED) as one of the areas or hotspots most affected by climate change on a global level [1]. Spain is located in this region. This country is already suffering from the impacts (social and economic losses) of the effects of climate change, particularly those related to meteorological phenomena, such as floods, droughts, heat and cold waves, sea storms and forest fires, among others.

Furthermore, Spain has a wide variety of climates within its territory, which shape different landscapes, depending on a series of physical elements, such as the geographic position, altitude, relief, proximity to the sea, the vegetation and fauna and flora of each territory.

In terms of water resources, this variety of climates implies a structural problem for Spain, in which two types of area can be distinguished: humid Spain, (the north-east, north and centre of the peninsula), which receives large annual amounts of precipitation with surplus water resources; and dry Spain (the east, south-east and south of the peninsula), where the average annual rainfall is very low, leading to deficits in the availability of water resources [2]. Paradoxically, some of the optimal lands for growing crops are found in parts of Spain with a water resource deficit, where the soil is rich and favourable for agricultural activities and where irrigated crops predominate over rain-fed crops [2].

The most important transfer in Spain constructed in the twentieth century is the Tagus–Segura Transfer (hereafter, TTS). This is the most important infrastructure given the volume of water it transfers, the areas it supplies and the political and media repercussions [2,3].

The introduction of the TTS has contributed a significant amount of water resources for both urban supply and for agriculture (irrigation) in south-east Spain. This contribution of water resources is considerable but insufficient, as it has only fulfilled the transfer volumes contemplated in the Preliminary Project of the Transfer once (water year 2000/2001) [4–6].

It is worth highlighting that in the 1990s (emergence of climate change hypothesis) to the present (2022) (complete confidence in climate change), the successive situations of rainfall droughts that affected the centre and south-east of the peninsula revealed that the TTS was vulnerable to extreme weather situations. In other words, the absence of rainfall in the Tagus headwaters for prolonged periods of time has a serious impact on the TTS [4–6]. Some authors calculated a period of NO transfers of water resources for approximately 15 months. Furthermore, these authors point out that by applying the same methodology, it could be said that the TTS would have been inoperative for a total of 59 months (5 years), from the hydrological year 2004/2005 to 2017/2018 [4].

As noted above, the TTS is vulnerable to extreme atmospheric situations (droughts). This problem is negatively aggravated considering the climate scenarios of emissions and effects on temperatures and precipitation in Spain [2,3,7–9], and, in specific terms, in the south-east peninsular region [10–12], from 2020 to 2050 and from 2015 to 2100, especially in river basins such as the Tagus and Segura.

Therefore, the relationship between the effects of climate change and water resources deserves special attention, given that the variations in climate that are occurring on a global scale are generating a series of effects in the Spanish territory, which is supported by rigorous scientific data. Some of the most concerning effects are the variation in atmospheric dynamics [12]; the increase in temperatures and the variations in rainfall [7,10–13]; the increase in temperature and the increase in sea level [14–18].

All of this has direct repercussions on water planning, given that the headwaters and resources in the basins are highly important for the development of agricultural activity, as they favour the accumulation of water resources in reservoirs and aquifers. Future water plans (third water planning period (2022–2027) and those of the following decades) should simultaneously contemplate solutions to address the reduction in the volumes of useful rainwater and the occurrence (ever more frequent) of intense or torrential rains leading to floods that cause increasing economic damage [10].

Within this context, one of the principal ways to obtain water resources which are not subject to climate variations is by increasing the so-called non-conventional sources, particularly wastewater treatment and desalination.

The former is subject to the water consumption of the population in a year. In this respect, there is a directly proportional relationship: the more water consumed by the population, the greater the availability of treated water, and vice versa. Regenerated water undoubtedly constitutes a buffer for the water resources of the basin and has been incorporated in the current water plan, acting as a complementary source to the resources of the TTS. However, from the end of the twentieth century and the beginning of the twenty-first century, the need to reuse treated water has arisen, giving rise to the passing of Royal Decree 1620/2007 of 7 December, establishing the legal framework for the reuse of treated wastewater, thereby promoting the development of the reuse of treated water and incorporating it into the water resources plan, provided that public health and environmental protection can be guaranteed and establishing the necessary requirements to enable or prohibit the use of treated or regenerated water, according to the afore-mentioned regulation [19,20].

Similarly to treated wastewater, desalinated water does not depend on climate variation. It only depends on its own daily and annual production capacity. Desalination is playing an increasingly prominent role in the hydrological planning of the basins. In Spain, the promotion of desalination began with the repeal of the Ebro Transfer project, through the passing of Royal Decree Law 2/2004 of 18 June, which modified Law 10/2001 of 5 July of the National Hydrological Plan and the implementation of the A.G.U.A. Programme,



and subsequently with the passing of Law 11/2005 of 22 June which modified Law 10/2001 of 5 July of the National Hydrological Plan [4,21].

This led to the planning and construction of large desalination plants along the Spanish Mediterranean coast. The largest desalination plant is that of Torrevieja. It has a current capacity of 80 hm$^3$/year and is managed by a state entity (ACUAMED). It has the largest capacity in Spain and one of the largest in Europe [22].

Desalination is characterised by being a strategic source and has been used in situations of severe drought in Spain, cushioning the effects of drought and providing a complementary water resource to the water of the TTS [4,5]. In fact, in situations of severe drought and when no transfers have been carried out, the desalination plant of Torrevieja has operated at full capacity, substituting the role of the transfer, for urban supply and irrigation [4,5].

While the rest of Europe uses desalinated water basically for urban supply, Spain is the pioneer in the use of desalinated water for agriculture and irrigation, given the water scarcity of the region [8,22].

Desalination in Spain has emerged in response to the transfer policy promoted by the former hydrological policy [5,6], which does not meet today's sustainability objectives. The desalination is accepted (socially) by the different economic sectors, and in many territories of the Spanish Mediterranean coast it has become a principal resource.

In short, the commitment to managing demand and the use of resources in a way that does not generate a situation in which territories are eternally dependent in terms of water is an essential and irreversible process [23]. It is necessary to break away from the traditional paradigm, based on the continuous supply of resources, which has no place in a scenario of climate change with less rainfall and a reduction in groundwater resources [23]. The growing use of "non-conventional" water resources will become a need in the coming decades on the Spanish Mediterranean coast, within the paradigm of demand and the sustainable use of water [23] (Figure 1).

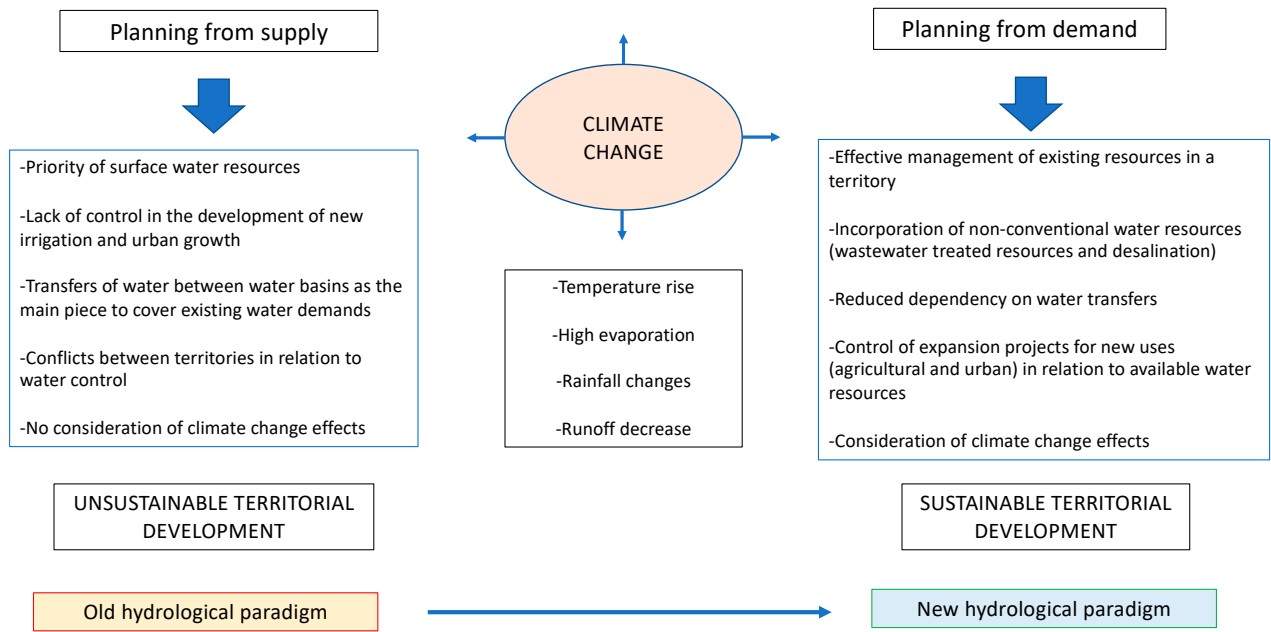

**Figure 1.** Scheme of the old and new paradigm in water planning in study area. Source: own elaboration.

The working hypothesis of the research is because the Tagus–Segura water transfer is being affected by the effects of climate change, especially regarding the quantity of water resources. To verify this hypothesis, a series of secondary objectives were set to corroborate this approach:

(a)    To ascertain the water situation in the headwaters of the Tagus basin.

(b)    To analyse the water balance and relevance of the TTS and non-conventional sources in the Segura basin, and their effect on the Lower Segura region.

(c)    To drawup a water balance adjusted to the Bajo Segura region to identify the agricultural areas with the greatest water demands.

(d)    To propose the basis for a new sustainable hydrological plan based on demand management and the use of own resources that are compromised by the foreseeable effects of climate change in terms of precipitation [24].

## 2. Materials and Methods

### 2.1. Background to the Tagus–Segura Transfer (TTS)

Beforehand, it should be put into context that the origin of the Tagus–Segura Transfer project dates back to the First National Hydraulic Works Plan (1933), which basically sought to correct the imbalances between the Atlantic and Mediterranean coasts which, through the so-called "Plan de Mejora y Ampliación de los Riegos de Levante" (Extension and Improvement Irrigation Plan in Spanish Levante region), was based on the transformation of a total area of 338,000 hectares, over the provinces of Murcia, Valencia, Alicante, Almería, Albacete and Cuenca [25].

After the severe drought of 1967, the TTS project was approved in 1969, the works were completed, and the diversion started operating in 1979. The diversion is a canal with a length of 286 km and a flow rate of 33 m$^3$/s. It links the Bolarque reservoir, in the Tagus basin, with the Talave reservoir, on the river Mundo, the main tributary of the Segura [25]. The cost of the construction of the diversion and post-transfer systems was estimated at ESP 90,000 million (La Verdad newspaper, 18 February 1998), equivalent to EUR 540,910,984 today.

According to the General Proposal for the Joint Management of the Water Resources of Central and South-eastern Spain, Tajo–Segura System, the final objective was to transfer an annual volume of 1000 hm$^3$. Of this, 640 hm$^3$/year would be used for irrigation. This objective would be met in two phases: a first phase, with a transfer of 600 hm$^3$, and a second phase, with a transfer of 40 hm$^3$. With these estimated volumes, it was expected to transform a total of 90,000 new hectares and complete the allocations of 46,816 existing deficit hectares. The latter were already under cultivation but did not have sufficient volumes of water for optimal irrigation. This was to be solved with the arrival of transferred water.

These planned volumes of water generated a great expectation that led to the transformation of rain-fed crops into irrigated crops (new irrigation), and the area benefited increased to 135,361 hectares. The area contemplated in the TTS project was 136,816, so that some authors indicate that "miraculously, it seemed that the objectives outlined in the preliminary draft had been achieved" [25]. However, the expansion of the surface area occurred during the years of construction of the TTS. Therefore, the increase in irrigated area was only justified using groundwater (indigenous resources) of the territories [25]. This implies that, for example, as in the case of the Vega Baja del Segura, most of the aquifers are overexploited. Consequently, the exploitation of the groundwater resources of the region, which are protected by the official basin organisation (Demarcación Hidrográfica del Segura), is currently prohibited.

Furthermore, this explains why the water from the aqueduct has been insufficient to supply the demand of the Segura basin, as indicated in the respective hydrological plans (2015–2021) and (2022–2027) [26,27], as the water from the aqueduct has been able to maintain, as far as possible, part of the transformed areas. Moreover, to make the best possible use of the water received from the aqueduct for new irrigation, modern irrigation techniques have been introduced, such as drip irrigation.

An interesting point in the Preliminary Project of the Tagus–Segura Transfer is that the demand in the Tagus basin was 1447 hm$^3$/year and the own adjustable resources amounted to 8152 hm$^3$/year, while the demand in the Segura basin was 1045 hm$^3$/year

and the available resources (basin) amounted to 820 hm³/year, revealing the deficit of the basin.

With the construction of the aqueduct, demand in the Tagus basin would be maintained, although the basin's resources would decrease by 1000 hm³/year due to the transfer of water from one basin to another, estimated at 7152 hm³/year. As for the Segura basin, demand would remain the same, but the available basin resources would increase thanks to the volumes of water transferred amounting to 2120 hm³/year (offer in the Segura basin). These figures have never been reached, as the Preliminary Project did not contemplate the possibility of an increase in demand in both basins (Tagus and Segura) or the absence of some transfers, for months or even years. In fact, the values estimated in the Preliminary Project have never been obtained with the transferred water.

### 2.2. Area of Study

Given that the research is focused on the Tagus–Segura water transfer, the chosen study area is divided into three parts. These three parts coincide with the presentation of the sections in the Results section (Figure 2).

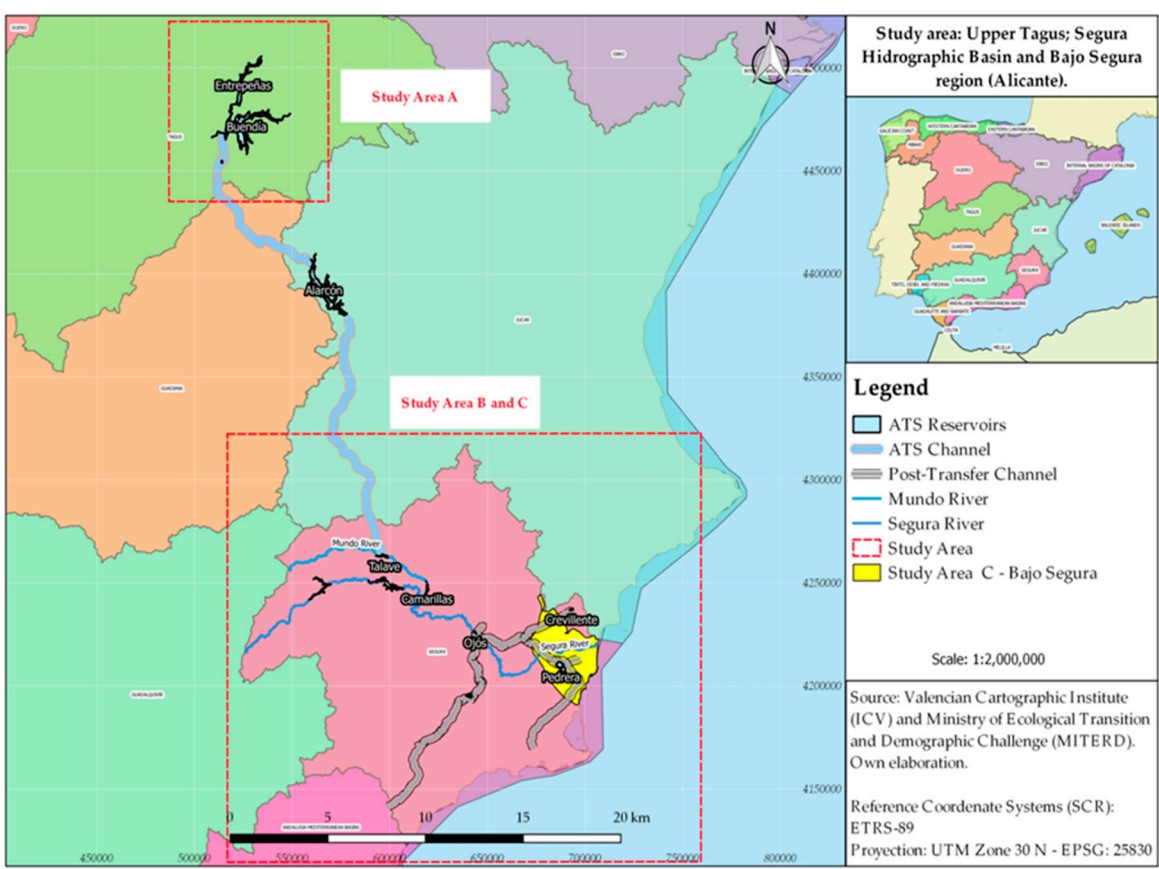

**Figure 2.** Study area: (**A**) Upper Tagus; (**B**) Segura Hidrological Basin; and (**C**) Bajo Segura Region (Alicante). Source: Valencian Cartographic Institute (ICV) and Demarcación Hidrográficadel Tajo (DHT) and Segura (DHS). Own elaboration.

The first area of analysis focuses on the river Tagus basin, specifically in the headwaters of the river Tagus, belonging to the sub-basin known as the Upper Tagus. The source of the river Tagus, the presence of two large reservoirs (Entrepeñas and Buendía) and the beginning of the hydraulic infrastructure of the Tagus–Segura Transfer are in this area (Figure 2). The effects of climate change calculated and estimated by the official basin organisation itself are also considered, in order to indicate the behaviour or trend in this sector.

The second zone corresponds to the Segura catchment area. The aspects analysed are those corresponding to the water balance (supply and demand) to ascertain whether the water transfer has made it possible to eliminate the existing deficit in the Segura basin, as was proposed in the Preliminary Project for the water transfer. Once the water balance is known, special attention is paid to the area corresponding to the province of Alicante. To this end, the UDAs (Agricultural Demand Units) corresponding to this region were selected. The purpose of this analysis is to find out the amount of existing gross or irrigable surface area, and to compare it with the net or irrigated surface area in this region, according to the data provided by the new Hydrological Planning Cycle of the Segura River Basin (2022–2027).The gross and net demand of the previously selected UDAs is analysed below. Gross and net demand is directly related to gross and net surface areas. Therefore, the demand makes it possible to know the volume of water necessary to supply these areas to obtain an optimal irrigation for the crops (calculated by the official basin organisation).

Lastly, the effects of climate change in the Segura basin are analysed, calculated by the basin organisation with respect to rainfall, evapotranspiration, surface runoff and aquifer recharge; this provides information on the water future of the Segura basin (which is structurally deficient).

The third area of analysis is centred on the province of Alicante (Valencian Community), specifically in the region known as Bajo Segura or Vega Baja del Segura. The choice of this area is justified by the fact that it is a region directly dependent on the water resources of the Tagus–Segura water transfer.

In this respect, when the water from the aqueduct reaches the Ojós reservoir, three water diversion channels start from this reservoir, which correspond to the so-called "Post-Transfer Infrastructure". These three channels take different directions. The first heads towards the province of Alicante, passing through the north of the district of Vega Baja del Segura, as far as the Crevillente reservoir (BajoVinalopó district and the Júcar basin). The second channel heads towards the Pedrera reservoir (Bajo Segura district). From this point, another canal splits into two branches: one heading north-east towards the town of Los Montesinos; and another heading south, passing through towns such as San Miguel de Salinas, Orihuela and Pilar de la Horadada (Bajo Segura district). This channel ends when it reaches the municipality of Cartagena (Murcia). The third canal heads southwards, crossing the Murcia Huerta, Algeciras, among other localities; it ends in Almería (Andalusia).

Given the great length of the water transfer and its implications for large areas of land, it has been decided to limit the analysis to the province of Alicante, specifically to the Bajo Segura or Vega Baja del Segura region (made up of 27 municipalities).

The territory of the district of Vega Baja is included within the Segura basin in terms of hydrological planning but is influenced by the Tagus basin as it receives resources from it through the Tagus–Segura Transfer. This reveals the need to analyse the current situation of each basin in relation to the effects of climate change that are visible in Spain, and their impact on the available water resources.

Along these lines, the district of Vega Baja del Segura implements a multi-source system in water management: surface water (River Segura), groundwater, the Tagus–Segura Transfer, the post-transfer, the reuse of treated wastewater and desalination. This situation is ideal in areas with a natural scarcity of water resources [24]. However, the system is vulnerable to the effects of climate change, due to the reduction in surface water resources predicted for this part of Spain. Therefore, it is necessary to reflect on possible future solutions to guarantee supply in this territory with natural rainfall scarcity.

### 2.3. Data Source and Analysis

This article analyses the deficiencies of the Tajo–Segura Transfer and its implications in the Segura River basin, and, especially, in the region of Bajo Segura or Vega Baja del Segura, motivated by the effects of climate change.

The methodology applied is based on the hypothetical-deductive model. The hypothetical-deductive method is one of the most accepted methods currently in the scientific field,

especially applied to social sciences such as geography, the approach given in this article. The method consists of a working hypothesis that, based on the analysis of a series of available data, allows the hypothesis to be corroborated or not. The steps of the hypothetical-deductive method are: (1) data collection; (2) data evaluation; (3) hypothesis generation; (4) diagnosis; and (5) final conclusion and/or proposal.

The working hypothesis of the research is based on the fact that the Tagus–Segura water transfer is being affected by the effects of climate change, especially with regard to the quantity of water resources. In order to confirm these aspects, information and data have been compiled from the institutions, bodies and official associations (Figure 3).

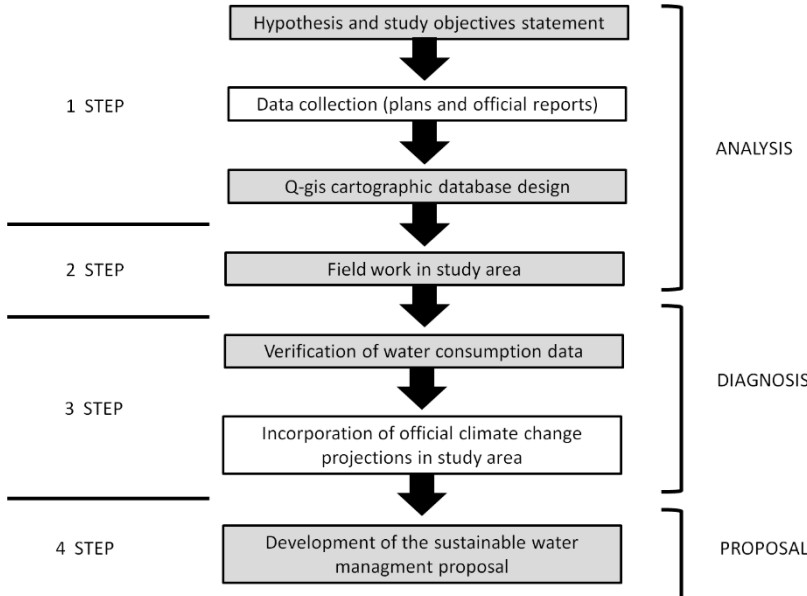

**Figure 3.** Diagram/outline of the methodological steps in this paper. Source: own elaboration. The innovative contributions of this paper are shown in the shaded areas.

First, the Hydrological Plan of the Tagus Basin (2022–2027) was consulted to compile information regarding the available water resources of the basin. Particular focus was placed on precipitation, surface runoff and the volume of water stored in the Entrepeñas and Buendía reservoirs. The predicted effects of climate change on the Tagus basin were also identified, specifically in the headwaters or sub-basin of the Tagus.

To obtain the data of the volumes of water stored in the Entrepeñas and Buendía reservoirs individually and jointly, the website of the Ministry for the Ecological Transition and the Demographic Challenge (MITERD) through the Centre for Public Works Studies and Experimentation (CEDEX) was consulted. This organisation has all the yearbooks of the volumes of all the basins in Spain, by day and month, depending on the time series chosen. In this way, the years of operation of the TTS have been established as a time series (1979–2021). The principal element in the search was to identify the reserve figure in hm$^3$ at the beginning of each month in order to analyse the evolution of the water stored in each reservoir or jointly (Entrepeñas–Buendía), so as to establish whether there are currently sufficient volumes of water to transfer via the TTS to the irrigated lands of south-east Spain, and whether these volumes are similar to the theoretical transfer volumes established in the Preliminary Project of the TTS.

After ascertaining the situation in the Tagus headwaters with respect to the precipitation, surface runoff and volume of reservoir-stored water, information and data were collected regarding the Hydrological Plan of the Segura Basin (1st Cycle 2015–2021) and the Hydrological Plan of the Segura Basin (2nd Cycle, 2022–2027).

In the case of wastewater treatment, the data were provided by the Regional Department of Agriculture, Rural Development, Climate Emergency and Energy Transition based

on the data of the Entidad Pública de Saneamiento de Aguas Residuales (Public Wastewater Treatment Entity) (EPSAR) of the Region of Valencia, which refer to the wastewater treatment plants (EDARs) installed in the district, their daily production capacity, their annual production capacity, the water treated and the water regenerated, among other information of interest.

With respect to desalination, data on production capacity and the water produced over the last four to five years were provided by the Torrevieja desalination plant. The public entity ACUAMED provided the data on the Torrevieja desalination plant and the cost of production, the energy cost and the future expansion projects contemplated for the desalination plant.

The different declarations of drought and the adaptation mechanisms adopted were also consulted for the period 2015–2018 to determine how the price of desalinated water produced by the Torrevieja plant was established at EUR 0.30 m$^3$ and the measures adopted by the basin in response to an extreme atmospheric event.

An analysis of these statements reveals a fundamental aspect that is significant for the future of TTS and traditional irrigation models in relation to water sources and extreme events. Therefore, it appears that HTM-dependent irrigation models enter a situation of pre-warning of drought long before the traditional irrigation models that use the basin's resources. In addition, traditional irrigation models do not enter a drought early warning situation until rainfalls fall to half of the drought early warning values in TTS crops. This justifies greater adaptation to extreme weather phenomena, such as droughts, and therefore resource-dependent irrigated crops.

Finally, in relation to the TTS and the volumes of water assigned and consumed in each irrigation community, the study establishes whether with this volume of water profits have increased or decreased, depending on the greater or lesser amount of available water. To do this, four key socio-economic indicators were taken into account: the area of production, the average budget per inhabitant, recruitment in agriculture and unemployment in agriculture for the district of Vega Baja del Segura.

In this respect, the volumes of water transferred in 2011, 2015 and 2019 were also included, together with the desalinated water produced by the Torrevieja plant (ACUAMED) to establish a relationship between the availability of water and the previously mentioned socio-economic indicators.

A geographical information system called QGIS was used for the cartographic part. The layers used to elaborate the location map of the area of study were downloaded in shape (shp.) format from the Valencian Cartographic Institute (ICV), the National Geographic Institute (IGN), the Demarcación Hidrográfica del Segura (DHS), the Demarcación Hidrográfica del Tagus (DHT) and the Demarcación Hidrográfica del Júcar (DHJ) (Table 1).

This analysis seeks to determine the water balance and to estimate the deficit that exists. It also proposes alternative measures to increase the water supply and a new hydraulic plan for the hydrographic basins of Spain.

**Table 1.** Sources and official documents consulted.

| Source | Documents Consulted | Information Used |
|---|---|---|
| **Spanish Meteorological Agency (AEMET)** | - Iberian Climate Atlas (2011)<br>- https://www.aemet.es/documentos/es/conocermas/recursos_en_linea/publicaciones_y_estudios/publicaciones/Atlas-climatologico/Atlas.pdf (accessed on 13 June 2022) | - Monthly and annual rainfall cartographic data of the study area (Tagus basin and Upper Tagus sub-basin). |

**Table 1.** *Cont.*

| Source | Documents Consulted | Information Used |
|---|---|---|
| **Tagus Hidrográfica Demarcación** | - Hydrological Plan for the Tagus River Basin (2009–2015).<br>- https://www.chtajo.es/LaCuenca/Planes/PlanHidrologico/Planif_2009-2015/Paginas/default.aspx (accessed on 13 June 2022)<br>- Hydrological Plan for the Tagus River Basin (2015–2021).<br>- http://www.chtajo.es/LaCuenca/Planes/PlanHidrologico/Planif_2015-2021/Paginas/Plan_2015-2021.aspx (accessed on 13 June 2022)<br>- Hydrological Plan for the Tagus River Basin (2022–2027).<br>- http://www.chtajo.es/LaCuenca/Planes/PlanHidrologico/Planif_2021-2027/Paginas/BorradorPHT_2021-2027.aspx (accessed on 13 June 2022) | - Average precipitation in the Upper Tagus sub-basin.<br>- Average runoff in the Upper Tagus sub-basin. |
| **Ministerio para la Transición Ecológica y Reto Demográfico (MITERD)** | - Yearbooks of gauges in Spain (1979–2021)<br>- https://ceh.cedex.es/anuarioaforos/afo/embalse-datos.asp?ref_ceh=3006 (accessed on 13 June 2022) | - Water stored in Entrepeñas and Buendía, and all of it, from 1979 to 2021. |
| **Demarcación Hidrográfica del Segura** | - Hydrological Plan for the Segura River Basin (2015–2021).<br>- https://www.chsegura.es/es/cuenca/planificacion/planificacion-2015-2021/plan-hidrologico-2015-2021/ (accessed on 13 June 2022)<br>- Hydrological Plan for the Segura River Basin (2022–2027).<br>- https://www.chsegura.es/es/cuenca/planificacion/planificacion-2022-2027/el-proceso-de-elaboracion/ (accessed on 13 June 2022)<br>- Actual consumption of the irrigation communities dependent on the TTS (2020) (data provided by CHS). | - Water balance (supply and demand).<br>- Supply resources.<br>- Resource demand.<br>- Urban demand.<br>- Agriculturaldemand.<br>-<br><br>Climatechangescenarios. |
| **Sindicato Central de Regantes del Acueducto Tajo Segura (SCRATS)** | - Volumes of water transferred for irrigation in the Segura basin.<br>- https://www.scrats.es/ (accessed on 13 June 2020)<br>- Water volumes corresponding to Alicante of the water transferred for irrigation in the basin (data provided by SCRATS). | - Volumes of water transferred for irrigation in the Segura basin.<br>- Volumes of water corresponding to Alicante of the water transferred for irrigation in the basin. |

**Table 1.** *Cont.*

| Source | Documents Consulted | Information Used |
|---|---|---|
| **Public Entity for Wastewater Sanitation (EPSAR)** | - Information on the existing wastewater treatment plants (EDARs) in the region of Vega Baja del Segura.<br>- https://www.epsar.gva.es/estaciones-depuradoras (accessed on 13 June 2020) | - Number of WWTPs installed in the Vega Baja del Segura.<br>- Number of WWTPs in operation in the Vega Baja del Segura.<br>- Design production capacity.<br>- Flows treated per day ($m^3/s$).<br>- Flows treated per year ($hm^3/year$).<br>- Flows treated and regenerated.<br>- Discharge points. |
| **Mediterraneanba- sinwaters (ACUAMED)** | - Current data from the Torrevieja plant<br>- (data provided by ACUAMED).<br>- Future expansion projects<br>- (data provided by SCRATS). | - Maximum desalination capacity at the Torrevieja plant.<br>- Energy cost of desalination.<br>- Production cost of desalination.<br>- Final cost of production and delivery.<br>- Environmental costs.<br>- Future planning (expansion of production capacity by $80 > 120 > 160\ hm^3/year$). |

Source: own elaboration.

The scientific contribution of this article is based on the analysis of the most current data offered by basin organisms, the effects of climate change and its involvement in water resources. First, it is shown that the current water situation at the head of the Tagus is not the same as in 1969. This implies a reduction in water resources in the Tagus basin and, consequently, lower volumes of water transferred. These data are demonstrated by the analysis of a short series of precipitation and runoff (1940–1979 and 1980–2018). To complete this analysis, we obtained the existing data of water embalmed in Entrepeñas and Buendía (separately) of the series of operation of the Tajo–Segura Transfer (1979–2020). In order to proceed with a transfer, it is necessary that the volume of water in the reservoirs of Entrepeñas–Buendía (as a whole) exceeds a certain amount at the beginning of each month.

The analysis of the water volumes packed in these reservoirs, both individually and separately, shows a significant reduction in the water resources stored, caused by the effects of climate change due to the lack of rainfall in the headwaters or the volume of rainfall. One of the effects demonstrated by the climate change that occurs in Spain is based on the reduction in rainfall in inland areas as opposed to large discharges that occur on the coast. This has a serious impact on the basins' water resources, since if it does not rain at the head of the rivers, the basins' water resources are diminished.

The analysis of the data from the Segura basin has allowed us to know the water balance (offer and demand) and the existing deficit, despite the water contributions of the Tajo–Segura Transfer. The data obtained have allowed us to know the situation in the province of Alicante, adjusting it to the region of Bajo Segura.

Another new aspect of the work corresponds to the construction of a water balance (supply and demand) and its water deficit, adjusted to the region of Bajo Segura in Alicante. The data show that the existing deficit is not as high as the above claims and that there is water for irrigation in the region. In addition, a table was prepared to estimate the situation for future scenarios in the Vega Baja of the Segura River (2030 and 2050). This table has been compiled as follows: the calculation of these quantities was established on the basis of the following method of analysis: (1) analysis of the official water demand and supply data calculated by the CHS; (2) analysis of the official data on the effect of the decrease in rainfall on water resources (AEMET, CEDEX) in the study area, especially of the resources coming from surface water (Segura River and Tajo–Segura water transfer); (3) knowledge of the territorial dynamics of the study area (for calculating future demands of the different water uses), which is what the field work was used for; (4) calculation of substitution flows for surface water and water transfer with non-conventional water (reuse and desalination), knowing that the reuse of wastewater has a ceiling (depending on the recent evolution of treated flows and total urban expenditure) and that the great asset is desalination for urban and agricultural use. Furthermore, in the current energy transfer process, it is estimated that the final cost of desalinated water will decrease in the coming years, depending on the installation of solar energy sources to feed the Torrevieja plant.

The most relevant of this analysis are two specific issues: (a) it is identified that the highest amount of water demand for irrigation cultivation corresponds to the trans-formed areas from rain-fed to irrigated and is dependent on the waters of the Tagus–Segura Transfer. It is also observed that irrigated crops dependent on basin water resources have a greater resistance to extreme weather (droughts) than those dependent on the transfer. In addition, it has been shown on numerous occasions that the transfer is an infrastructure vulnerable to such situations. (b) The data provided by the basin body include the volume of concessional water allocated to each irrigation community dependent on TTS water in the province of Alicante. However, actual consumption data by irrigation communities are much lower than the volume of water allocated in 1969.

The fact that they do not receive the amounts of water allocated in 1969 is justified by the reduction in water available at the head of the Tagus, hence the importance of having made its analysis earlier. The data show that these irrigation communities will never receive such volumes of water allocated in 1969, when the climate and water reality was quite different from the current one (2022).

However, farmers in the region continue to think that they do not receive such amounts of water for political-social reasons, forming the so-called "water wars". Examples of this are exaggerated claims that it is "the end of Europe's market garden", "without water there is not agriculture" or "that there is a deficit of 1000 hm$^3$/year in the Segura basin". Farmers are therefore still waiting for water that they will never receive again, hence the importance of this article to demonstrate the real situation with rigorous scientific data.

The last aspect to be highlighted is that the deficit in the region of Bajo Segura can be solved by increasing the volume of water from wastewater treatment and desalination, increasing its annual production capacity, as proposed by the basin body. There are also other methods of reducing the deficit, such as the use of modern irrigation systems or the reduction in less productive, if extremist, cultivated land.

All these issues justify the scientific and novel contributions of the article.

## 3. Results

### 3.1. The Effects of Climate Change in the Headwaters of the Tagus (Upper Tagus)

According to the Hydrological Plan of the Tagus Basin (2022–2027), the current average rainfall in all Spanish areas of Tagus basin is 594 mm (1980–2018 series). These data are something different in the Upper-Tagus sub-basin, where the great reservoirs that regulate the Tagus–Segura Transfer are located and have experienced a decrease between the 1940–79series (655 mm.) and the 1980–2018 series (568.5 mm.).Therefore, now the average rainfall is lower than the average rainfall for the entire Tagus basin.

Many studies highlight the reduction in rainfall in the headwaters of the Tagus basin, calculated at a decrease of 12% in the period 1980–2018 [8,9]. This justifies the Tagus basin plan with the rainfall data (average and maximum) and surface runoff (average and maximum) for the sub-basin of the Upper Tagus, the headwaters of the Tagus and the beginning of the Tagus–Segura Transfer [28].

It should first be clarified that there are three series in the Hydrological Plan of the Tagus Basin: 1940–2018 (long series), 1940–1980 (old short series) and 1980–2018 (current short series). Only the last two short series were considered, using the mean values. The headwater rainfall was calculated from the SIMPA model, which is the hydrological reference model for the calculation of water resources in the hydrological plans. For the calculation of rainfall, the model establishes average data for each planning area within the basin, and when there are no long series in the meteorological observatories of the area, it fills in data from the longest series corresponding to observatories close to the planning area.

The results clearly show the effects of climate change in the headwaters of the Tagus basin in relation to precipitation. Although in some months there is an increase, the general monthly trend corresponds to a considerable decrease. In general terms, it can be observed that the mean obtained in the short series (1940–1979) was 655 mm and in the most recent short series (1980–2018) the mean drops sharply to 568.5 mm of precipitation in the headwaters of the Tagus. This represents a decrease of -86.5 mm of precipitation, which represents a decrease of 13.2%, at present (Table 2). Consequently, these means are substantially different, which implies evidence of the impact of climate change in the Tagus headwaters [28].

**Table 2.** Calculations of average monthly/annual precipitation decrease and percentages in the Upper Tagus (series 1940–1979 and 1980–2018).

| | Series 1940–1979 (mm) | Series 1980–2018 (mm) | Reduced Quantity (mm) | Percentage Reduction (%) |
|---|---|---|---|---|
| October | 29.2 | 30.3 | +1.1 | +3.8 |
| November | 31.3 | 32.3 | +1 | +3.2 |
| December | 47 | 53.2 | +6.2 | +13 |
| January | 80.1 | 67.6 | −12.5 | −15.6 |
| February | 90.4 | 64 | −26.4 | −29 |
| March | 91.2 | 68.2 | −23 | −25 |
| April | 79.7 | 68.8 | −10.9 | −14 |
| May | 68.1 | 61.2 | −6.9 | −10 |
| June | 45.1 | 44.4 | −0.7 | −1.6 |
| July | 36.5 | 31.9 | −4.6 | −13 |
| August | 27.6 | 24.7 | −2.9 | −11 |
| September | 28.8 | 21.9 | −6.9 | −24 |
| **Total** | **655** | **568.5** | **−86.5** | **−13.2** |

Source: Hydrological Plans for the Tagus Basin (2022–2027; 2015–2021; 2009–15). Monthly collated data *Iberian Climate Atlas* (AEMET, 2011). Own elaboration. Color: Identify decreases (red and negative) and increases (green and positive).

Meanwhile, the Hydrological Plan of the Tagus Basin (2022–2027) also reports a series of data related to surface runoff. In this respect, the reduction in rainfall has a direct effect on the surface runoff of the Tagus Basin. These volumes of surface water are used for the transfer.

Therefore, surface runoff functions as an indicator of the impact of climate change, in relation to the reduction in mean precipitation in the headwaters of the Tagus. Adding the monthly averages of surface runoff for the short series (1940–1979) gives a total surface

runoff of 657.4 hm$^3$ for the 30-year series. For its part, the sum of the monthly averages of surface runoff for the short series (1980–2018) gives a total result of 380.8 hm$^3$ in the 30-year series. This implies the reduction of a total of 276.6 hm$^3$ in thirty years, which is a percentage reduction of 42.1% (Table 3) [28].

**Table 3.** Calculations of average monthly/annual runoff decrease and percentages in the Upper Tagus (series 1940–1979 and 1980–2018).

|  | Series 1940–1979 (Hm$^3$) | Series 1980–2018 (Hm$^3$) | Reduced Quantity (Hm$^3$) | Percentage Reduction (%) |
|---|---|---|---|---|
| October | 30.9 | 24.9 | −6 | −19.4 |
| November | 44.3 | 30.6 | −13.7 | −31 |
| December | 55.9 | 58.4 | +2.5 | +4 |
| January | 84.1 | 48.4 | −35.7 | −42.4 |
| February | 130 | 45.8 | −84.2 | −65 |
| March | 157 | 71.7 | −85.3 | −55 |
| April | 79.6 | 49.9 | −29.7 | −37.3 |
| May | 50.3 | 35.8 | −14.5 | −28.8 |
| June | 11.0 | 11.7 | +0.7 | +6 |
| July | 1.0 | 0.4 | −0.6 | −60 |
| August | 0.5 | 0.5 | 0 | 0 |
| September | 12.8 | 2.6 | −10.2 | −80 |
| **Total** | **657.4** | **380.8** | **−276.6** | **−42.1** |

Source: Hydrological Plan for the Tagus Basin (2022–2027). Own elaboration.

Another aspect that highlights the decrease in rainfall and surface runoff in the Upper Tagus due to climate change is the volume of water stored in the Entrepeñas and Buendía reservoirs, particularly when analysing the historical series of 1979–2020, coinciding with the years of the operation of the TTS.

In the water year 1979–1980, the volume of water stored in the Entrepeñas reservoir reached a value of 6308 hm$^3$/year. This figure had fallen to 4083 hm$^3$/year in the water year 2019–2020. Meanwhile, in the Buendía reservoir, in the year 1979–1980, there was a total volume of 14,268 hm$^3$/year, and in the water year 2019–2020 this figure had fallen sharply to 3613 hm$^3$/year. However, it should be remembered that the volume of water authorised for transfer depends on the sum of the two reservoirs (Entrepeñas and Buendía). In this respect, and following the same line of analysis as in the individual cases, the total joint volume of water of the Entrepeñas and Buendía reservoirs in water year 1979–1980 amounted to 20,576 hm$^3$/year, with 7696hm$^3$/year in water year 2019–2020, representing a decrease of 63% (Table 4, Figure 4).

Moreover, it should be noted that in the years of severe drought, as a whole, values of between 5000 and 9000 hm$^3$/year have been reached, revealing their vulnerability to extreme atmospheric events.

Therefore, two conclusions may be drawn: (a) the impact of the effects of climate change on the headwaters of the Tagus is inevitable, as a reduction in rainfall directly affects the surface runoff and the volumes of water stored in reservoirs; and (b) the TTS is adversely affected in this respect, given that the theoretical volumes calculated to transfer to south-east Spain correspond to volumes of water that existed in and prior to 1979–1980, which are not attainable in the present day. This implies the need to carry out a review of the calculations made in the Preliminary Project of the Transfer with current data, given that it is inconceivable that the theoretical water allocations can continue to be planned based on calculations made over forty years ago, when the climate reality was completely different to that of today.

**Table 4.** Comparison between water stored in the Entrepeñas–Buendía reservoir and volumes of water transferred by the Tagus–Segura Aqueduct.

| Hidrologycal Year | Vol. Water (hm³/year) | Vol. of Water Transferred (hm³/year) |
|---|---|---|
| 1979–1980 | 20,576 | 36 |
| 1980–1981 | 15,308 | 463.9 |
| 1981–1982 | 9648 | 287.9 |
| 1982–1983 | 5853 | 287.9 |
| 1983–1984 | 5142 | 287.9 |
| 1984–1985 | 9674 | 373.6 |
| 1985–1986 | 10,431 | 386.4 |
| 1986–1987 | 10,129 | 307.2 |
| 1987–1988 | 10,574 | 436.9 |
| 1988–1989 | 11,086 | 438.6 |
| 1989–1990 | 9679 | 287.9 |
| 1990–1991 | 8628 | 287.9 |
| 1991–1992 | 6646 | 287.9 |
| 1992–1993 | 5307 | 287.9 |
| 1993–1994 | 5025 | 255.1 |
| 1994–1995 | 3321 | 150.8 |
| 1995–1996 | 5383 | 361.1 |
| 1996–1997 | 11,882 | 457 |
| 1997–1998 | 15,964 | 302.5 |
| 1998–1999 | 14,229 | 287.9 |
| 1999–2000 | 9945 | 287.9 |
| 2000–2001 | 11,798 | 373.6 |
| 2001–2002 | 10,090 | 287.9 |
| 2002–2003 | 9276 | 312.9 |
| 2003–2004 | 9486 | 382.7 |
| 2004–2005 | 7673 | 287.9 |
| 2005–2006 | 3635 | 287.9 |
| 2006–2007 | 4139 | 253.3 |
| 2007–2008 | 3918 | 138.7 |
| 2008–2009 | 5298 | 199.2 |
| 2009–2010 | 9818 | 259.2 |
| 2010–2011 | 14,202 | 287.8 |
| 2011–2012 | 11,169 | 287.8 |
| 2012–2013 | 8926 | 287.8 |
| 2013–2014 | 9048 | 287.8 |
| 2014–2015 | 6236 | 281.2 |
| 2015–2016 | 5396 | 231.8 |
| 2016–2017 | 4637 | 127.3 |
| 2017–2018 | 5224 | 156.6 |
| 2018–2019 | 7297 | 313.6 |
| 2019–2020 | 7696 | 210 |

Source: yearbooks of gauges in Spain. Own elaboration.

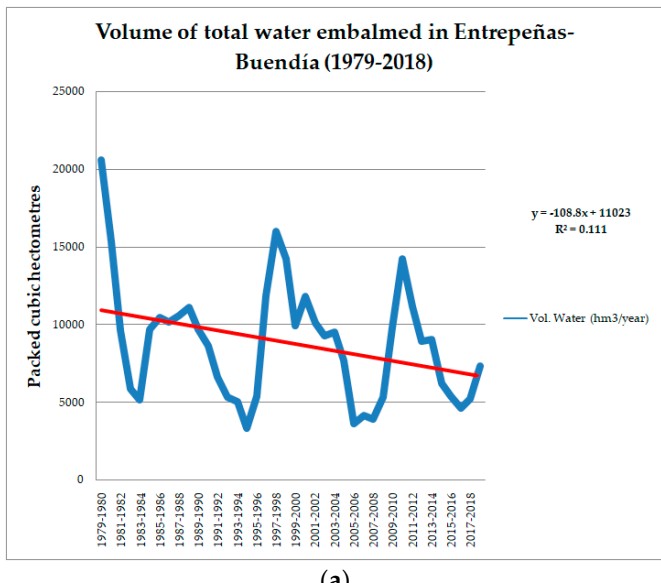
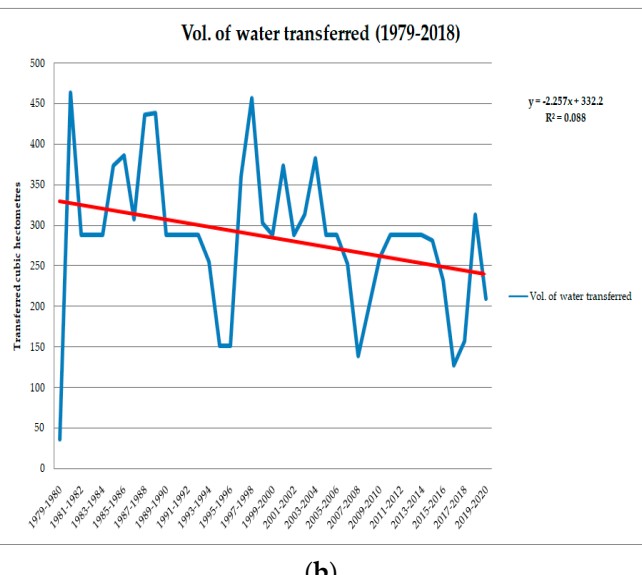

(**a**)　　　　　　　　　　　　　　　　(**b**)

**Figure 4.** Graph comparison between water stored in the Entrepeñas–Buendía reservoir and volumes of water transferred by the Tagus–Segura aqueduct. (**a**) Evolution of water reservoirs in the Entrepeñas–Buendía reservoir (1979–2018); (**b**) evolution of volumes of water transferred by the Tagus–Segura aqueduct (1979–2018). Source: yearbooks of gauges in Spain. Own elaboration.

Finally, the PHCT (2022–2027) reports the percentage of change in the quarterly runoff, calculated in average (RCP 4.5) and high (RCP 8.5) climate change scenarios for the Upper Tagus. For the months of October to December, a reduction of 14% is calculated in an RCP 4.5 scenario and 20% in an RCP 8.5 scenario, coinciding with the months of most rainfall [28]. This implies that the rainfall trend, the surface runoff and the volumes of reservoir-stored water in the Upper Tagus are in continuous decline, constituting a serious problem for those dependent on the water from the TTS.

*3.2. The Water Balance in the Segura Basin: Supply and Demand in Vega Baja*

According to the Hydrological Plan of the Segura Basin (2015–2021), the total available resources of the basin (including the TTS) amounted to 1511hm$^3$/year, and the total demand of the basin reached a volume of 1841 hm$^3$/year. Given that the demand is higher than the available resources, the water deficit calculated for the whole of the Segura basin for this time horizon was 330 hm$^3$/year [26].

Meanwhile, the recent Hydrological Plan of the Segura Basin (2022–2027) indicates that the total available resources in the basin (including the TTS) amount to 1571 hm$^3$/year, which implies an increase in water resources in the basin. The total demand of the basin, meanwhile, amounts to a volume of 1831 hm$^3$/year, representing a decrease in demand with respect to the previous time horizon, which is justified by the increase in water resources from non-conventional sources. This is whythe deficit has reduced to 260 hm$^3$/year (Table 5) [27].

Considering the scenarios contemplated for 2027 and 2039 in the PHCS (2022–2027), two different trends may be identified: the first is the continued increase in water resources using non-conventional sources, such as those obtained through wastewater treatment and desalination in the Segura basin for 2027 and 2039. The second trend is related to the Segura basin's own water resources, such as surface and groundwater, which are set to decrease in terms of total annual volume. Furthermore, the plan considers that less sea discharges will take place in 2027 and 2039, which suggests an advance in the use and management of water.

A rising trend can be observed in the urban demand in the Segura basin. This coincides with the increase in agricultural demand for the 2027 and 2039 horizons [27]. This

constitutes a serious problem for agriculture, given that the Water Law 29–1985 and the subsequent Law of the National Hydrological Plan 2001 and Law 11/2005, which modifies the previously mentioned law, indicate urban supply as the priority use. This implies that in situations of extreme atmospheric events (droughts), priority is given to urban supply above other uses (ecological, irrigation and agricultural uses, industrial, etc.). Therefore, in the case of need, water will be extracted from the water allocations assigned to irrigation and agricultural users, thereby aggravating the consequences for irrigated crops.

**Table 5.** Segura River basin water balance (2022–2027).

| Segura River Basin Water Balance | | Average Resources (hm$^3$/year) Horizon (2022–2027) |
|---|---|---|
| **Available Resources** | Surface water | 704 |
| | Groundwater | 66 |
| | Non-draining surface resources to the Segura | 15 |
| | Returns to the system | 268 |
| | Desalination | 223 |
| | TTS | 295 |
| | **TOTAL AVAILABLE RESOURCES** | **1571** |
| **Demand** | Urban demand | 250 |
| | Environmental demand | 39 |
| | Agricultural demand | 1522 |
| | Other demands (industrial, golf, etc/. | 20 |
| | **TOTAL BASIN DEMAND** | **1831** |
| | **BASIN DEFICIT** | **−260** |

Source: Hydrological Plan for the Segura River Basin (2022–2027).

With respect to the urban demand of the basin, the calculation of urban demand conducted by the CHS in the district of Vega Baja del Segura was performed. In 2021, urban demand was recorded at 39.2 hm$^3$/year, and it is estimated that by 2039 it will have increased to 43.5 hm$^3$/year [27]. It is interesting to note that the urban demand corresponding to the district of Vega Baja only represents 16% of the total demand of the Segura basin.

It is more interesting to analyse the agricultural demand of the district of Vega Baja del Segura, according to the data included in the PHCS. To achieve this, the gross or usable agricultural land and the net areas or those cultivated by UDAs (Units of Agricultural Demand) corresponding to the district of Vega Baja del Segura were chosen (Table 6).

As we can observe in Table 5, the PHCS shows that there is a gross or usable agricultural area of 65,411 hectares, of which 47,636 hectares are cultivated, representing 72.8% [27]. In the afore-mentioned plan, the total gross and net demands per crop and the total demand for per UDA are reported.

Table 5 shows the UDAs that depend on the waters of the Tajo–Segura Transfer are 52, 53, 56 and 72. This gives a total of 27,837hectares and an annual water demand of 117.9 hm$^3$/year, while the UDAs that depend on available basin resources (46, 48, 51 and 55) add up to a total of 19,799 hectares and an annual water demand of 76,5 hm$^3$/year. These data show that the UDAs that were rain-fed crops and were transformed into irrigated crops (new irrigation), because of the Tajo–Segura Transfer project, are the areas with the highest water demand and, consequently, the causes of the water deficit in the region of Bajo Segura.

**Table 6.** Area (gross and net) and demand (gross and net) by UDAs belonging to the region of Bajo Segura (Alicante).

| | | Gross Area (ha) | Gross Demand (hm³/year) | Net Area (ha) | Net Demand (hm³/year) |
|---|---|---|---|---|---|
| **UDA** | **Denomination** | Horizon 2022–2027 | | Horizon 2022–2027 | |
| 46 | Traditional Vega Baja | 23,780 | 100.1 | 15,469 | 58 |
| 48 | Vega Baja, post. Al 33 y ampl. del 53 | 3067 | 12.3 | 1913 | 8.2 |
| 51 | Mixed irrigation of aquifers and wastewater treatment plants south of Alicante | 4538 | 9.9 | 1634 | 7.5 |
| 52 | Riegos Levante Right Bank | 3439 | 15.9 | 2886 | 11.9 |
| 53 | Riegos Redotados del TTS de RLMI-Segura | 11,046 | 52.4 | 8713 | 36.1 |
| 55 | Crevillente Aquifer | 1306 | 3.2 | 783 | 2.8 |
| 56 | La Pedrera ZRT TTS Redotados Irrigations | 10,563 | 52.5 | 9411 | 41.4 |
| 72 | Re-dedicated irrigated lands of the Vega Baja TTS, left bank | 7672 | 40 | 6827 | 28.5 |
| | **TOTAL** | **65,411** | **286.3** | **47,636** | **194.4** |

Source: PHCS (2022–2027).

It is evident that climate change has a significant impact on water, which obliges the organisations of the basin to take it into account in hydrological planning. In this respect, the CHS establishes three future periods of 30 years, called impact periods (IP). These impact periods are IP1 (2010–2040), IP2 (2040–2070) and IP3 (2070–2100), reflecting the impact in the short, medium, and long term, in accordance with the medium (RCP 4.5) and high (RCP 8.5) climate scenarios (Table 7) [27].

**Table 7.** Effect of climate change with respect to an unaffected situation on hydrological variables in the DHS.

| | | Med RCP 4.5 | Med RCP 8.5 |
|---|---|---|---|
| **Precipitation** | **PI (2010–2040)** | −2% | −5% |
| | **PI (2040–2070)** | −4% | −10% |
| | **PI (2070–2100)** | −8% | −14% |
| **Real evapotranspiration** | **PI (2010–2040)** | −2% | −5% |
| | **PI (2040–2070)** | −4% | −9% |
| | **PI (2070–2100)** | −6% | −11% |
| **Recharge** | **PI (2010–2040)** | −7% | −10% |
| | **PI (2040–2070)** | −12% | −23% |
| | **PI (2070–2100)** | −20% | −36% |
| **Runoff** | **PI (2010–2040)** | −7% | −9% |
| | **PI (2040–2070)** | −11% | −23% |
| | **PI (2070–2100)** | −20% | −38% |

Source: PHCS (2022–2027).

In fact, the short-, medium- and long-term impacts of climate change related to water resources reveal a negative scenario for the Segura basin. As we can observe in Table 6, there will be a reduction in rainfall of between 8% and 14% in the Segura basin by the

end of the century. The potential and real evapotranspiration translate into negative effects; the humidity of the soil will also decrease; the aquifer recharge will face serious problems, oscillating between 20% and 36%; and finally, the surface runoff will decrease sharply throughout the whole of the Segura basin, with values fluctuating between 20% and 38% [27].The reason for using the RCP 8.5 scenario, which is characteristic of an extreme scenario of high emissions, is justified because the trend in emissions is increasing annually, the effect of climate change on the reduction inwater resources is notorious and, finally, because it justifies the proposal put forward in this research to carry out sustainable water planning.

*3.3. The Water Balance in the District of Vega Baja del Segura and the TTS*

Obtaining data on water resources (supply) in the region of Vega Baja del Segura has been a complex task. This is because there is no previous study of the water balance in this region. The data found are only represented in two scales: river basin scale or provincial scale (+140 municipalities, Alicante).

For this reason, the data that represent the water balance of the region of Bajo Segura are adjusted to its political limit composed of the 27 municipalities that make up it.In this sense, the data have been found in different bibliographies of official bodies such as the Segura River basin or the Diputación de Alicante, among others. All these data have been adjusted for the regional scale of work (Table 8).

**Table 8.** Water balance 2021 in the region of Vega Baja del Segura (Alicante).

| WaterBalance in the Region of Vega Baja del Segura (Alicante) | | Average Resources (hm$^3$/year) |
|---|---|---|
| **Offer** | Surface water | 40 |
| | Groundwater | 36 |
| | Returns to thesystems | 32 |
| | TTS | 61.1 |
| | Wastewater treatment | 25 |
| | Desalination | 48.1 |
| | **TOTAL DISTRICT RESOURCES** | **242.2** |
| **Demand** | Urban demand | 39.2 |
| | Environmental demand | 32 |
| | Agricultural demand | 194.4 |
| | Other demands (industrial, golf, etc.) | 8 |
| | **TOTAL DISTRICT DEMAND** | **273.6** |
| | **DISTRICT DEFICIT** | **−31.4** |

Source: own elaboration based on PHCS (2022–2027), SCRATS (2020), EPSAR, ACUAMED, Generalitat Valenciana and Diputación de Alicante data.

The estimated deficit in the district of Vega Baja del Segura (Alicante) is 31.4 hm$^3$/year. It should be noted that urban demand is completely guaranteed by the water resources from the River Taibilla, the TTS and the desalinated water used by the Mancomunidad de los Canales del Taibilla, which supply the municipalities of Vega Baja del Segura. Meanwhile, environmental demand is also guaranteed by the circulating waters of the river Segura and the returns to the system, which are used as ecological flows in the final section of the river Segura. The other demands (industrial, golf, etc.) are covered using water purified with a tertiary or advanced treatment. The problem of the deficit resides in agricultural demand, particularly in the so-called new irrigated lands, because the waters from the transfer are insufficient to supply the existing demand (Table 9).

**Table 9.** Estimated situation for future scenarios in the Vega Baja of Segura River (2030 and 2050).

| Water Resources Vega Baja | 2021 | 2030 | 2050 |
|---|---|---|---|
| Surface water | 40 | 38.7 | 34 |
| Groundwater | 36 | 30 | 20 |
| Returns to system | 32 | 33.1 | 35 |
| TTS | 61.1 | 57.4 | 52.1 |
| Wastewater treatment | 25 | 26.4 | 29 |
| * Desalination | 48.1 | 80 | 120 |
| **Total Resources** | **242.2** | **265.6** | **290.1** |
| Urban demand | 39.2 | 41.4 | 45 |
| Environmental demand | 32 | 32 | 32 |
| Agricultural demand | 194.4 | 194.4 | 176.9 |
| Other demands (industrial, golf, etc.) | 8 | 8 | 8 |
| **Total Demand** | **273.6** | **275.8** | **261.9** |
| **Deficit/Surplus Vega Baja** | **−31.4** | **−10.2** | **+28.2** |

* Desalination: by 2030 the Torrevieja desalination plant will be expanded to 120 hm$^3$/year, of which 80 hm$^3$/year will be for agricultural use and 40 for urban use. By the year 2050, it is estimated that the production capacity will be 160 hm$^3$/year, of which 120 hm$^3$/year will be used for irrigation and the remaining 40 hm$^3$/year for urban use. Source: own elaboration based on the scenarios proposed in the Segura River basin and the medium climate scenario RCP 4.5.

Of the 194.4 hm$^3$/year of agricultural demand, according to the PHCS (2022–2027), a total of 160.8 hm$^3$/year correspond to UDAs 52, 53, 56 and 72, which coincide with the sectors of the new irrigated land (former rain-fed land transformed into irrigated land); hence, there is high demand for water for two reasons: (a) the lands were originally dry and modern irrigation systems allow irrigated crops to be grown on them, and (b) the farms or plots are large with areas similar to those of latifundios (large estates).Therefore, a large amount of water is necessary to maintain their productivity and enable these areas to continue to produce irrigated crops (Table 10).

**Table 10.** Agricultural demand that takes advantage of the waters of the TTS in the Vega Baja del Segura.

| Denomination | Agricultural Demand hm$^3$/year (2021–2027) |
|---|---|
| UDA—52. Segura RLMD | 15 |
| UDA—53. Segura RLMI | 52 |
| UDA—56. Redotated irrigation of the TTS of the La Pedrera ZRT | 52 |
| UDA—72. Vega Baja, left bank redotated irrigation systems | 40 |
| **TOTAL** | **160.8** |

Source: PHCS (2022–2027). Own elaboration.

In this respect, the Confederación Hidrográfica del Segura provided a series of data referring to the volume supplied of the transfer water to the 29 irrigation communities in the water year 2019/2020 (Table 11). This table reflects two relevant aspects: the concessional volume and consumption. The concessional volume refers to the amount of water assigned to each of the irrigation communities that are beneficiaries of the transfer water, corresponding to the theoretical values in the Preliminary Project of the Transfer, given that the total sum of the concessional volumes assigned to each irrigation community amounts to 123 hm$^3$/year, 30% of which is allocated to the province of Alicante. It would be very difficult to achieve these theoretical amounts in the current climate and water context and considering the effects of climate change.

**Table 11.** Volume of water supplied by the water transfer to the CC.RR. of Alicante in the hydrological year 2019–2020.

| | Transfer Resources | |
|---|---|---|
| | Concesional Volume (m$^3$) E (m$^3$) | Consumption (m$^3$) |
| C.R. Albatera | 7,815,324 | 3,969,496 |
| C.R. Las Cuevas | 1,491,100 | 796,865 |
| T.D. Lo Belmonte | 666,925 | 356,416 |
| T.D. Lo Marques | 485,366 | 259,386 |
| T.D. Las Majadas | 767,010 | 409,903 |
| C.R. San Joaquin | 479,950 | 241,840 |
| C.R. Santo Domingo (Grupo 3.490) | 2,276,500 | 1,216,595 |
| C.R. Lo Reche | 1,473,892 | 731,114 |
| C.R. El Carmen | 571,739 | 305,301 |
| T.D. Manachon Candela | 111,000 | 29,380 |
| C.R. Riegos de Levante (M.I.) | 77,512,272 | 26,213,062 |
| C.R. Riegos de Levante (M.D.) | 5,500,000 | 1,299,612 |
| C.R. La Fuensanta Grupo 2000 | 1,007,750 | 206,895 |
| C.R. La Estafeta | 55,100 | 20,963 |
| C.R. Murada Norte | 2,001,700 | 1,069,736 |
| C.R. El Mojon | 1,156,641 | 385,076 |
| C.R. Perpetuo Socorro | 1,709,400 | 816,432 |
| C.R. Las Dehesas | 961,350 | 477,008 |
| C.R. Barranco de Hurchillo | 239,250 | 127,858 |
| C.R. San Onofre y Torremendo | 1,715,350 | 916,708 |
| Agrícolas Villamartin | 110,200 | 57,358 |
| C.R. Río Nacimiento | 627,850 | 335,532 |
| C.R. Mengoloma de Orihuela | 208,800 | 111,795 |
| C.R. Pilar de la Horadada | 2,621,600 | 1,401,019 |
| C.R. San Isidro y Realengo | 7,500,000 | 0 |
| C.R. Campo Salinas | 2,122,800 | 1,041,434 |
| C.R. San Miguel | 1,922,700 | 773,152 |
| C.R. Las Cañadas | 150,800 | 81,064 |
| **TOTAL** | **123,262,369** | **43,651,000** |

Source: DHS (2020).

It is interesting to know the consumption per irrigation community, given that, if the value of consumption of each irrigation community has allowed sufficient irrigation for the crops to grow correctly and produce yields that are profitable for the farmer, this implies that the net agricultural demand of the beneficiaries of the TTS water fluctuates between the consumption values of 2020, that is, around 44 hm$^3$/year (Table 11).

To corroborate this point, the data regarding the irrigated area of the district of Vega Baja del Segura provided by the Valencian Institute of Statistics (IVE) of the Regional Department of Sustainable Economy, Productive Sectors, Trade and Employment were consulted. These data reveal that there are variations in the size of the cultivated area in accordance with the droughts occurring and the availability of water resources. In this

respect, in the years 2014–2016 (years of drought in the Segura Basin), there was an increase in the irrigated area, with an area of 16,399 hectares recorded in 2014 and 16,955 hectares in 2016. In the following two years of drought (2017–2018), the irrigated area decreased to values fluctuating between 16,500 and 15,500 hectares. Finally, it should be noted that in 2019, the cultivated area increased again to 16,321 hectares, a value close to that of 2020.

From the data provided by the IVE, it may be concluded that the municipalities with irrigated crops dependent on the TTS waters (new irrigated land) reduced their cultivated area in drought situations, as in the case of the municipalities of Orihuela, Albatera, Benferri, Pilar de la Horadada or San Miguel de Salinas, among others. Meanwhile, the irrigated crops that depend on the water resources from the Segura basin (Segura River) increase or maintain the same areas of irrigated crops in drought situations, even in the years of most severe rainfall and hydrological drought in the basin, as in the case of the years 2012, 2015 and 2018.

The loss of irrigated area dependent on the TTS water resources could have been even greater in the years 2015–2016 and 2017–2018 (drought in the Tajo, Júcar and Segura basins, among others), if the desalination plant of Torrevieja had not started operations as a strategic source to support the TTS in this type of situation [5].

Meanwhile, the data provided on the ARGOS Information Portal of the Regional Government of Valencia allows the addition of a series of socio-economic indicators that enable the relationship existing between economic development (agriculture) and the greater or lesser availability of water to be evaluated (Table 12).

**Table 12.** Socio-economic indicators in the Vega Baja del Segura in relation to the greater or lesser availability of water resources.

| | 2011 | 2015 | 2019 |
|---|---|---|---|
| Regional cultivatedarea | 16,425 | 16,955 | 16,321 |
| Average budget per habitant | EUR 745.87 | EUR 844.30 | EUR 938.72 |
| Registered hiring in agriculture | 11.87% | 21.83% | 28.06% |
| Unemployment in agriculture | 4.22% | 6.34% | 4.59% |
| Annual transfer TTS | 85.9 | 32.8 | 85.5 |
| Desalination | N/D * | 30 | 76.4 |

* N/D: non data. Source: Valencian Cartographic Institute (ICV) (IVE), ARGOS and ACUAMED.

Thus, with respect to the average budget per inhabitant, a gradual increase from the year 2011 to 2019 may be observed. It continued to rise until 2021, reaching a value of EUR 1026.35/inhabitant.

Regarding the recruitment recorded in agriculture, in the year 2013–2014 an exponential increase in percentage terms can be observed, reaching a figure of 30.68%. This value decreased in the following year due to the severe drought (21.83%) and continued to decrease until 2018. From 2019, a growing trend began in the percentage of recruitment recorded in agriculture, reaching a value of 35.01% in 2021.

On the other hand, registered unemployment in agriculture displays different behaviour. From 2007 to 2016, unemployment in agriculture gradually increased. The highest values correspond to the drought years of 2015 and 2016, reaching 6.16% and 6.34%, respectively. From 2017 until the present day, the number of unemployed in agriculture has decreased to a level of 4.59 in the year 2020.

These indicators become more significant when they are examined in relation to the greater or lesser availability of water from both the TTS and desalination.

In March 2015, a drought situation was declared in the Segura basin, for which 12 exceptional measures were implemented for the management of water resources and EUR 30 million of extraordinary credit was assigned. In September of the same year, the

drought continued, and the drought declaration was extended until September 2016. In addition to the measures already implemented, further action was taken with respect to the control of the continental waters. In October 2015, for the first time in Spanish history and in the Segura basin, among the new measures announced by the Ministry was the reduction in the price of desalinated water produced by the Torrevieja plant to 0.30 EUR/m$^3$, with the authorisation of the production of 30 hm$^3$/year and a subsidy of EUR six million to reduce the cost of production [29].

This highlights that, thanks to the production of desalinated water in the years 2015 and 2016, the area of irrigated crops in the district of Vega Baja del Segura remained the same. This is also visible in a similar value maintained of total agricultural income in the Region of Valencia.

In March 2018, Law 1/2018 of 6 March was passed, referring to the adoption of urgent measures to mitigate the effects generated by the drought in certain hydrographic basins. Additionally, the Revised Text of the Water Law, approved by Royal Legislative Decree 1/2001 of 20 July was modified, in response to the continued drought situation in a large part of Spain. This explains the decrease in the cultivated area in the years 2017, 2018 and 2019 in the district of Vega Baja del Segura, together with the reduction in agricultural income. Another problem that explains this worsened situation is that, from the year 2016–2017, the price of desalinated water increased to 0.60–0.80 EUR/m$^3$. Farmers could not afford this price and opted to not cultivate or even abandon their irrigated land due to the absence of a guarantee of water resources.

However, in the water years 2017–2018, when there was a drought in the headwaters of the Tagus, the Torrevieja desalination plant reduced the price of its desalinated water (for the second time) to 0.30 EUR/m$^3$ for irrigators. This cost was farbelow that of production costs [5]. This coincided with the months of an absence of transferable resources or "no transfer", which obliged the desalination plant of Torrevieja to produce desalinated water at almost maximum capacity in 2019 [29,30]. The effects for the cultivated land and socio-economic aspects were positive, thanks to desalination.

## 4. Discussion

The results reveal a series of problems in hydrological planning in Spain which directly affect the district of Vega Baja del Segura, in relation to the Tagus–Segura Transfer.

The analysis shows that the volumes of water available in the sub-basin of the Upper Tagus are not the same as those of fifty years ago, given that the average and maximum rainfall have reduced, the surface runoff has decreased and the volumes of water independently stored in the Entrepeñas and Buendía reservoirs have decreased sharply. This behaviour has also been observed in the joint water resources of Entrepeñas–Buendía since the beginning of operations of the TTS.

The reduction in rainfall recorded in the last period of climate analysis (1980–2018 series) is 15% with respect to the 1940–1979series and is consistent with what the CEDEX points out in its report on the impact of climate change on water resources in Spain [8] and with what Marcos and Pulido point out [31]. A decrease in winter and spring rainfall is also observed, which are the most effective for hydrological planning purposes (urban tourist and agricultural demands in summer) and a slight increase in autumn rainfall, in line with what has been pointed out by various authors [32–35] for the eastern sector of the Iberian Peninsula. Thus, the monthly distribution of precipitation in the headwaters of the Tagus tends to be "Mediterranean", with a more prominent participation of autumn rains.

The climate trends and their effects related to water resources in the Tagus basin (Upper Tagus) with respect to rainfall, surface runoff and the volume of water stored in reservoirs reveal a progressive reduction in available water resources. Taking into account the climate change scenarios (RCP 4.5 and RCP 8.5) and even after undertaking adaptation tasks, the Tagus basin is experiencing serious problems, which are being demonstrated and tested with scientific data.

As mentioned above, the Mediterranean region is the planetary zone where the effects of climate change will be most devastating [1]. In this sense, there are numerous scientific publications that have analysed this issue for the Mediterranean region. These studies show that, although the reduction in precipitation is not linear, but rather amplified (due to the increase in extreme weather events), two types of behaviours have been observed in the Mediterranean region. The first of these is obviously the reduction in average precipitation and the form of rainfall [11]. Greater amounts than historical records can now fall in a very short time. The second is that rainfall tends to concentrate on the Mediterranean coast and not in inland areas, where the headwaters of Spain's main rivers (such as the Tagus and Segura Rivers) are located [33–37]. If there is no rainfall in the headwaters of these rivers, there will hardly be any water resources in the basin and, consequently, no transfer from one basin to another will be possible. Therefore, it is considered that the TTS and the current hydraulic planning is unsustainable in the context of climate change.

The problems of the Upper Tagus directly affect the hydrological planning of the Segura basin, through the Tagus–Segura Transfer. One of the problems identified in the period of operation of the TTS (1979–2021), except for one year, is that the theoretical volumes assigned in the Preliminary Project of the Tagus–Segura Transfer have not been achieved, not even under the initial operating regulations. In fact, with respect to the demand and available water resources in the Segura basin thanks to the transfer, the Segura basin has never reached more than 2000 $hm^3$/year as an available resource, even with the TTS, wastewater treatment and desalination. Meanwhile, demand has increased exponentially.

This is mainly due to the problems in the headwaters experienced in the Tagus basin because of climate change. The figure that demonstrates this aspect is the average volume of water transferred from the origin (208 $hm^3$) and received in the destination (182 $hm^3$), indicated by the CHS for the period contemplated [27]. Furthermore, the data of the SCRATS (Central Irrigation Syndicate of the Tagus–Segura Aqueduct) reveal that, of the 400 $hm^3$/year assigned to irrigation in south-east Spain (theoretical), the average volume of the transfer for the period indicated is 195.6 $hm^3$, that is, less than half of the assigned amount. Of the 400 $hm^3$/year (theoretical), 30% corresponds to the province of Alicante, which is 120–125 $hm^3$/year (theoretical). However, the reality is very different. Taking the data of the CHS into account, and particularly that of the SCRATS, the average volume of water received in the province of Alicante from the TTS is 61.1 $hm^3$ in the period of operation (1979–2021). This implies that the province of Alicante receives half of the theoretical amount of water assigned to it.

This raises the following questions: Why has no reassessment or review been made of the operating calculations of the TTS in over 50 years? Why is planning undertaken with values that do not currently exist, nor will exist in the future? Why do the farmers denounce and call for volumes of water that they will never receive? The answer is clear: they are being deceived because they have been given no explanation about the current situation of the hydrographic basins because of climate change.

Therefore, the TTS displays a series of significant weaknesses subject to the variations in climate, and the territories which are supplied with its water resources must begin to adopt measures that do not directly depend on it (self-sufficiency). To obtain a greater volume of water, desalination and wastewater treatment are being used to contribute to the supply, enabling the deficit of the basin to be reduced to some extent.

However, the trends in urban and agricultural demand are growing, with the latter having the highest demand. It should be remembered that in the case of drought or a shortage of urban water supply, the water used to supply the urban nuclei will be drawn from other uses, such as ecological flows or those used for agriculture. Therefore, agriculture is losing water resources, and in the medium to long term, if measures are not taken to correct these problems, it will be seriously affected.

In this respect, and as a proposal for an empirical and demonstrated adaptation, the new irrigated lands (dependent on the TTS) enter a situation of pre-alert of drought much before the traditional irrigated lands. This was evident in the drought of 2015, after the

passing of RD 356/2015 of 8 May, declaring the situation of drought in the territorial region of the Confederación Hidrográfica del Segura with the adoption of exceptional measures for the management of resources, due to the reduction in the interannual contributions (rainfall) in the headwaters of the Segura and Tagus basins and its successive extensions (RD 335/2016). In other words, the resources of the Segura basin can supply the traditional irrigated lands in situations of extreme drought, although not the crops dependent on external resources. This should constitute an incentive for changing the way hydrological planning is conducted in the Segura Basin, the Vega Baja district and in the rest of Spain.

With respect to the Alicante district of Vega Baja del Segura, the deficit existing is due to agricultural demand: principally, the irrigation communities that depend on the water resources of the transfer. This is because after the TTS project was approved, the construction of the transfer took 10 years until it began operation. During this period, there was much speculation in Vega Baja del Segura regarding the water that would be received, based on the afore-mentioned theoretical volumes. This speculation translated into the transformation of rain-fed crop land into irrigated land using groundwater. Therefore, the real use of the TTS has not increased the irrigated areas, but has maintained, as far as possible, all the transformed areas since then.

The current status of the non-conventional sources allows them to considerably reduce the deficit existing in Vega Baja del Segura, although this does not mean that there is no longer a deficit. If the desalination plant of Torrevieja operated at its maximum capacity ($80 \text{ hm}^3/\text{year}$), with the support of the treated wastewater ($24$–$25 \text{ hm}^3/\text{year}$), the deficit would be reduced to $10.5 \text{ hm}^3/\text{year}$. The deficit would be resolved with the extension of the maximum production capacity of the Torrevieja desalination plant from 80 to 120, and subsequently to $160 \text{ hm}^3/\text{year}$. This proposal is contemplated in the PHCS (2022–2027). In parallel, there would be an increase in the volume of reusable treated wastewater, which would complement the desalinated water. These two sources would become the principal water sources of the district of Vega Baja del Segura. The role of the TTS, therefore, would be a secondary or complementary source to these resources, when needed.

However, this increase in production should be accompanied by mechanisms imposed by the government based on Law 7/2021 of 2 May regarding climate change and energy transition, as a framework within which to reduce the cost of production of desalinated water, based on energy subsidies. In this way, the farmers would be provided with water for irrigation at a price of $0.20$–$0.30 \text{ EUR/m}^3$, as in the case of the two occasions when the TTS failed during situations of extreme drought.

Taking all these issues into account, Spain needs to review and reconsider the current hydrological policy. To do this, first, it should begin by reviewing the values of the Preliminary Project of the TTS and adjust them to the volumes of water currently available. Furthermore, the water contributions assigned to each irrigation community dependent on the water from the TTS should be reviewed, given that they do not reflect the current situation. This analysis will reveal the available water resources. Second, after determining the situation of the water resources related to the TTS, it would be appropriate to increase the maximum capacity of the Torrevieja desalination plant (ACUAMED) in order to obtain larger volumes of water resources and maintain the plant in full operation as well as obtaining a price of desalinated water that is affordable for farmers. Third, agricultural demand should be reduced using new irrigation systems (drip system), a switch to irrigated crops that require less water or a radical change from irrigated crops to rain-fed crops and, as a more extreme measure, the reduction incultivable areas.

The new hydrological policy in Spain should be constructed on sustainability, climate change scenarios, hydrologic planning and on measures of adapting to climate change in a horizon of 100 years.

The *new sustainable hydrological planning* should be based on scenarios of climate change elaborated by the IPCC in its reports and on its regional effects. Sustainable planning means that, despite the existing resources, this exercise should be contextualised within the worst climate scenario possible, that is, in RCP 8.5 scenarios. Planning a climate and water

resources situation based on the RCP 8.5 severity will enable a management of water resources able to guarantee the resource for the rest of the twenty-first century in Spain. If the reality is very different to that contemplated in the plan (due to a limitation of greenhouse gases, the implementation of measures to adapt to extreme atmospheric events and the respect and fulfilment of all the treaties and agreements in terms of reducing emissions), there will be a surplus and more water will be available for the assigned uses (urban supply, ecological flows, irrigation and agriculture, industry, leisure, tourism, etc.).On the contrary, if the scenarios contemplated in the RCP 8.5 are fulfilled, a prior adaptation to this situation will have already been contemplated and planned. Furthermore, the adoption of this approach would enable measures of adaptation to be developed over the years.

This new planning proposal will significantly slow down the environmental deterioration and socio-economic losses. If these aspects are not considered from today, the consequences will be much more severe in the medium and long term, and losses running to millions of euros will be incurred. It would be particularly severe for those crops dependent on the TTS. The implementation of hasty and drastic measures as the RCP 8.5 climate scenario approaches will lead to greater economic investments with dubious profitability. Furthermore, this new policy should be flexible and open to modifications which enable adaptations and adjustments to be made in accordance with the climate situation and the future scenarios contemplated.

The new planning and management scenario proposed in this paper is in line with the proposal in the mentioned Law 7/2021 on climate change in Spain, which advocates the incorporation of the effects of climate change into hydrological planning (art.19) to increase the resilience of the different uses of water. In particular, agricultural uses must adapt the demands to the expected resources to minimise the expected impact on future climate scenarios. The proposal for sustainable hydraulic planning in our study area is in correspondence with various authors in Spain who advocate a reduction in agricultural irrigated surfaces compared to the increase contemplated in various current hydrological demarcation plans (Guadalquivir, Guadiana, Tajo), or the maintenance of existing ones that are not sustainable, in present day and in climate change scenarios (Segura, Júcar) [36–38].

Moreover, the new sustainable hydraulic plan should focus on the management of the demand for water, with a commitment to non-conventional sources, desalination and wastewater treatment, with the afore-mentioned determinants (increase in capacity, reduction inthe production costs of desalinated water, mechanisms to reduce the energy and production costs of desalination, applying tertiary treatment to purified water so they may be reused for agricultural and other uses). After managing the resources of non-conventional sources, the water resources of the hydrographic basins should then be included (surface, groundwater, returns to the system, sea discharges, etc.). Next, the external water resources from other basins should be incorporated, such as those from the Tagus–Segura Aqueduct, as a strategic or complementary source which, when necessary, can transport water to satisfy the demands. Furthermore, it is also necessary to calculate the water needs per crop in order to determine the amount of water that is required to produce substantial yields of the crop and its fruit, administrating the necessary volume of water.

Another aspect that the new plan should contemplate isadaptation measures through water use agreements, such as those in Marina Baja (Alicante), where the irrigators concede clean water for urban supply while the regenerated water is used for irrigation in agriculture. To achieve this, it is preferable to establish an agreement between the interested parties (farmer, water company and the EDAR, among others).

After planning all these factors, it is necessary to supply all of the demands existing in the basin and territory as far as possible. Most likely, in the deficit basins which have transformed areas of rain-fed land into irrigated land (new irrigated land dependent on the TTS), different alternatives will have to be sought to satisfy their demand: desalinated water,

treated wastewater, new irrigation systems, changing to soils and crops that require less water, or, in the most extreme case, reducing the crop areas in order to bring down demand.

The proposal presented in this study adapts to the objectives established by the Spanish government for the year 2050 which seek to promote the development of alternative sources of supply (reuse and desalination based on renewable energy), reduce the water lost in the sanitation and supply network, increase the quality of the water, "renewable water" and moderate consumption, among other actions.

The energy consumption of these facilities is currently around 3 Kw/h for each $m^3$ of water produced (generally less than 4 in new facilities including auxiliary systems and other pumping) and has been reduced from values of over 20 Kw/h/$m^3$ in the 1960s to current values [39,40].

Values of more than 20 Kw/h/$m^3$ in the 1960s to have lowered to current values [39,40] thanks to improvements in the chemistry and configuration of the membranes and in the systems for recovering residual energy from the brine.

Energy consumption is the largest cost of desalinated water production, so its reduction is the key factor in reducing the price of desalinated water. The implementation in Spain of the National Integrated Energy and Climate Plan (2021–2030) and the EU legislative package "Fitfor 55" responds to the European Commission's recent commitment to reduce net greenhouse gas emissions by at least 55% by 2030.

In this context, projects are being developed for the implementation of solar farms to supply desalination facilities on the Spanish mainland and in the Canary Islands. In particular, the Torrevieja desalination plant, a key element in the supply of desalinated water, is developing a project to install an electrical substation, powered by solar energy, with the aim of obtaining self-consumption of energy. At present, the desalination plant's energy consumption is estimated at 264 GWh, which would increase to 400 GWh with the expansion of its desalination capacity to 120 $hm^3$/year (from the current 80 $hm^3$/year).

As mentioned above, the production capacity of the Torrevieja desalination plant is currently 80 $hm^3$/year, where 40 isfor urban supply and another 40 for irrigation in the Segura River basin. However, the CHS plans to increase this to 120 in the current Hydrological Basin Plan (2022–2027). According to information provided by ACUAMED, the current specific consumption value of the Torrevieja plant varies between 3.25 and 3.65 kWh/$m^3$, depending on the delivery point and the required water quality. It is capable of producing 1 $hm^3$ per day (24 h), obtaining water with a conductivity of 200 μS, i.e., with a quality identical to that of mineral water and, therefore, water that can be used for irrigation. The economic cost of desalination for the years 2019, 2020 and 2021, and the resulting average tariff over the last years, is approximately 0.45 EUR/$m^3$ (water delivered).

Finally, they are rigorously monitoring the marine ecosystem impact caused by the discharge of brine or brine overflow. In this case, they ensure that they have zero environmental impact, as before dumping the brine they mix it with seawater and dump it in small quantities and in different areas to avoid causing damage to the marine environment. The Torrevieja desalination plant has eight sensors that monitor the salinity level, which have never once detected an environmental problem.

To achieve these water-related objectives for the year 2050, among the several actions proposed, the sixth addresses the need to "*adjust the management of water resources, preparing the system that will prevail in a future with a lower availability of water*" [41]. Therefore, "*a comprehensive water management strategy must be designed that promotes reuse and the desalination of water until its price is competitive, that is, similar to the price of water from traditional sources; improve the efficiency of the systems of urban supply, agricultural irrigation and the treatment of drinking water and wastewater, through the modernisation of infrastructures and the introduction of new technologies; reorder the agricultural and crop uses, acting on the prevailing concessional regime, prioritising sustainable and socially fair agriculture; increase the resilience of farms to extreme atmospheric events and the effects of climate change, through the transformation of crops and production systems, improve training in agricultural management and create adequate financial and governance mechanisms; and, finally, implement an ambitious strategy for restoring the rivers,*

*aquifers and other continental aquatic systems, while strengthening the river reserves and other protected spaces*" [41].

In short, the objectives established for Spain in 2050 are aligned with the proposal of the new hydraulic planning proposed in this research.

## 5. Conclusions

The climate and water situation in Spain and its respective hydrographic basins, particularly the Tagus and Segura basins, is not the same as it was 50 years ago, when the volumes of water to transfer via the Tagus–Segura Transfer were planned.

The Tagus basin has suffered a significant reduction in rainfall, surface runoff and volumes of reservoir-stored water in the sub-basin of the Upper Tagus. This is the starting point of the Tagus–Segura Transfer, and the reduction in water resources (surpluses) available in the Tagus basin has been modified by the effects of climate change. This has given rise to a serious problem, given that there will be less and less resources available to transfer. Therefore, the scenario of the Tagus basin will be to plan its own resources in order to supply the needs of its own basin, without taking into account the Segura basin, which is dependent on the waters of the TTS.

This implies that the Segura basin and the irrigated lands of south-east Spain will receive a lower volume of water from the transfer than theycurrently receive, given that the surplus resources of the Tagus basin are subject to variations in climate, rendering the transfer an infrastructure vulnerable to the effects of climate change. This is why the Segura basin should begin to make a firm commitment to using non-conventional sources, such as treated wastewater and desalinated water, the latter being the most important for the self-sufficiency of the basin, focusing on three fundamental aspects: (a) increasing the production capacity, (b) reducing the cost of the desalinated water supplied to the farmers (0.20–0.30 EUR/m$^3$) and (c) reducing the environmental impacts (brine).

Furthermore, the farmers in south-east Spain, particularly those of the new irrigated lands dependent on the TTS, should know that they have been deceived with the promise of water resources assigned through the water contributions for each irrigation community, when these volumes were calculated more than 50 years ago when the climate reality was completely different. Furthermore, these volumes are theoretical and have never been fulfilled, at least in the province of Alicante. The undersupply of the TTS is not due to a failure to transfer the resources that should be transferred, but because there are not enough resources in the headwaters of the Tagus that can be transferred to fulfil the theoretical volumes.

Therefore, in response to the questions posed at the beginning of this article, it is clear that the Tagus–Segura Transfer is not the only solution for the water future of the district of Vega Baja del Segura. There are other alternative sources, such as treated wastewater and desalinated water, which can increase the volume of available resources to supply the urban and agricultural demands. However, it should be noted that, with the current maximum capacity for producing desalinated water in the district (80 hm$^3$/year), the demand in Vega Baja cannot be satisfied. Therefore, it is necessary to extend the desalination plant of Torrevieja, increasing the production capacity to 120 hm$^3$/year and complemented by the regenerated volumes of water. In this respect, it is necessary to extend the treatment plants of the district of Vega Baja del Segura with tertiary or advanced treatment, given that most of them only perform secondary treatment and the resulting water cannot be applied directly to the crops.

Finally, Spain should be more ambitious in terms of hydrological planning. It is incomprehensible that the plan that is in force with respect to the TTS is based on the theoretical volumes of a Preliminary Project when the climate situation was very different to that of the present day. No attempt has been made to reconsider the operating water volumes for the current scenario taking into account the effects of climate change.

For this reason, to correct the lack of a coherent and rational water plan that contemplates the climate reality and effects of future climate change, this study proposes a new sustainable hydraulic plan.

This new sustainable plan requires a profound restructuring of hydrologic planning, based on the worst climate scenario (RCP 8.5), enabling a plan to be elaborated with a long-term horizon (until the end of the century) and adapted to climate change. This scenario will lead to significant restrictions with respect to the current water allocations, whereby the assigned volumes of water will be initially reduced or eliminated. The positive side of this plan is that if the reality in terms of climate evolves over the years into a scenario with less emissions and a better adaptation and management of water, the volumes of water assigned to each use may be increased. This is the main reason why the principal sources in the sustainable hydraulic plan are treated and desalinated water, with the aforementioned improvements, as these resources do not depend on climate variations. Then, the basin resources will be included. As observed in the severe droughts occurring in Spain, the basin resources, and, therefore, the traditional irrigated lands that use this water, are more resilient and are adapted to extreme atmospheric conditions, giving them a clear advantage for the effects of climate change. Subsequently, the external water resources will be included, which, depending on the climate evolution over the coming years, will be based on the transfer or not of resources from the Tagus basin to the Segura basin. Therefore, in this proposal, instead of playing a principal role, the TTS is used as a strategic-secondary source to support the principal sources in the proposed plan. Moreover, the uses, crops, areas cultivated and water allocations will have to be reordered in order to reduce demand and, therefore, the deficit existing in the Segura basin in Vega Baja.

The new sustainable hydraulic plan is committed to the self-sufficiency of the territory and only in case of need would it request external resources. However, for a territory to be resilient to the future effects of climate change, it is necessary to start acting now with respect to the afore-mentioned aspects to minimise the socio-economic losses (job positions, crops, desertification, land, cultivated areas, significant economic losses in terms of production and agricultural income, among many others). All of this is possible through a logical plan and coherent actions. A failure to take such measures would result in consequences that will be catastrophic for south-east Spain and, particularly, for the farmers and their way of life. This proposal establishes the fundamental pillars for a long-term national strategy for Spain in the year 2050.

**Author Contributions:** Formal analysis, investigation, resources, data curation, writing, A.O.C.; formal analysis, investigation, resources, data curation, writing, J.O.C.; formal analysis, data curation, supervision, writing, C.J.B.C. All authors have read and agreed to the published version of the manuscript.

**Funding:** This research received no external funding.

**Data Availability Statement:** The data can be found in the documents referenced in this article. Likewise, the authors offer to provide any data requested by other researchers or interested parties.

**Conflicts of Interest:** The authors declare no conflict of interest.

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
