# Peer review of "The Effects of Climate Change on the Tagus–Segura Transfer: Diagnosis of the Water Balance in the Vega Baja del Segura (Alicante, Spain)"

_water, doi:10.3390/w14132023_

Round 1
Reviewer 1 Report
The manuscript by Antonio Oliva et al reports “The Effects of Climate Change on the Tagus-Segura Transfer (TTS): diagnosis of the Water Balance in the Vega Baja del Segura (Alicante, Spain)”. This article is based on a collection of facts and information from government agencies responsible for water management (States, Hydrographic Confederations, previous research, and personal contributions), which has been synthesized and analyzed to provide a better understanding. A new sustainable water strategy is suggested, based on water resource planning according to availability and territorial self-sufficiency through the use of non-conventional sources like desalinated water and treated wastewater.
I still believe that this paper could be cited as a "review" better than an article, it is not only about the novel elements, it is about also the length. Overall, this work has been improved; however, still there are some issues that deserve clarification and should be carried out. Below are some issues and ideas for the author's consideration.
First of all, I would like to thank the ACADEMIC EDITOR for his efficient comments which has a very efficient role in improving this paper.
-Comment 1: Please remove any abbreviations from the title of the paper, “TTS”.
-Comment 2: The title of the figures should be written under the figures, not above them.
-Comment 3: I don't find the sentence 'Own elaboration' under figures and tables comfortable for the readers please fix that.
-Comment 4: please check the references carefully.
-Comment 5: The table in figure 4 is not readable, please reinsert it.
-Comment 6: Please follow the journal guideline for the table form and style.
Reviewer 2 Report
The authors responded satisfactorily to my comments. However, in Tables 2 and 3, authors should clearly indicate the means that are significantly different.
Author Response
Please see the attachment

This manuscript is a resubmission of an earlier submission. The following is a list of the peer review reports and author responses from that submission.
Round 1
Reviewer 1 Report
Evaluation report of the manuscript
The manuscript presents interesting results and is overall well written. However, there are improvements to be made before it is released.
Introduction
- The introduction is relatively long. You have to shorten it.
- The introductory text starting from line 158 and ending at line 200 (including figure 1) should be transferred to the methodological part.
Materials and Methods
- The use of survey methods (interviews) and GIS analysis is not clearly justified. Moreover, the results obtained by these methods are not presented in the Results section.
- The internet addresses of the organizations must be added to allow readers to possibly check the data.
Results
Text that starts from line 321 and ends at line 393 (including figure 2) should be transferred to the methodological part
Point 3.1. : Tables 2 and 3Authors should compare data from the periods 1940-1979 with those from 1980-2018 to demonstrate the differences between these two periods. Data from 1940-2018 should not be used to determine these differences. In addition, authors should apply an adequate statistical test to determine these differences.
Table 7The RCP8.5 table is an extreme and quasi-hypothetical scenario. The authors must justify the choice of this scenario.

Reviewer 2 Report
The manuscript by Antonio Oliva et al reports “The Effects of Climate Change on the Tagus-Segura transfer: diagnosis of the Water Balance in the Vega Baja del Segura (Alicante)”. This article is based on a collection of facts and information from government agencies responsible for water management (States, Hydrographic Confederations, previous research, and personal contributions), which has been synthesized and analyzed to provide a better understanding. A new sustainable water strategy is suggested, based on water resource planning according to availability and territorial self-sufficiency through the use of non-conventional sources like desalinated water and treated wastewater.
I saw this paper type being cited as an "article" but it is closer to a review. Overall, this work is very interesting; however, some issues deserve clarification and should be carried out, which are not discussed detailed-wise. I recommend a minor revision for this manuscript. Below are some issues and ideas for the author's consideration.
Comment 1: The English language needs some attention as some sentences are not quite correct and clear. You can find the grammatically corrected manuscript in the attached file.
Comment 2: The important research findings must be highlighted in the abstract. The general routine writing must be removed and should be highlighted with research findings.
Comment 3: The last keywords (wastewater treatment and Vega Baja del Segura) is not suitable for an article, it is too long and not in keyword form; write only ‘wastewater’.
Comment 4: figures are in very low quality, please improve their quality or redraw them.
Comment 5: What is the novelty of this work? There is so much literature already available on the Effect of Climate Change on the Tagus-Segura. So how this work is different from other reported work in literature. Here is an example of a similar paper type about Desalination, a strategic and controversial resource in Spain (June 2017WIT Transactions on Ecology and the Environment 216:61-72 DOI: 10.2495/WS170061).
Comment 6: Interpretation of search results should be improved, please if it is possible add more discussion, especially about the new sustainable water policy that was proposed.
Reviewer 3 Report
Review "The Effects of Climate Change on the Tagus-Segura transfer: diagnosis of the Water Balance in the Vega Baja del Segura (Alicante)" by Oliva et al.
Comments
L158-200 should go in M&M section. Also improve quality of Fig1
L224-226 please consider stating the novelty and contribution of this work
Section 2, M&M is too cryptic. It needs to be streamlined to explicitly state the multiple lines of evidence authors are trying to analyze. Also consider adding a flowchart/diagram depicting the methodological steps
L321-393, this is contextual information about the ATS, hardly any analysis of results here. Also consider improving Fig 2
Tables 2 & 3, it would be appropriate to report the decrease in rainfall and runoff with a certain significance level
L446-455. this assumes both time series (long-term & short-term) are representative and explain climate change in the study area.
L490-491, There is no data supporting this statement in the article.
L536. Table 7, these estimates need some sort of uncertainty assessment and/or statistical significance discussion. Probably the most important comment here is the fact that impacts across the water balance from reductions in rainfall are non-linear (in fact, they are amplified)
L725-728, these are conveyance losses, how are they explained by climate change?
L738-748, while data seems to support decreases in available water resources, this section includes some strong statements.
Recommendation
Work presented in this article is closer to a factual account of multiple data sources depicting the hydrological landscape of the Vega Baja del Segura. The science value of this work is limited and the novelty and contribution regarding the current state-of-the-art is questionable. I am inclined to reject this article in its present form.
I would recommend the authors to re-frame the work around a diagnosis of the hydrological situation in the case study, then find issues arising from that diagnosis, and then propose specific measures addressing those issues.
Round 2
Reviewer 1 Report
The authors have generally responded satisfactorily to my comments. Nevertheless, there are still minor corrections they need to make to their manuscript before publication.
Corrections to be made
- Point 2.2. (objectives) must be included in the Introduction part.
- The title of point 2.3. should change to “data sources and analysis”.
- For the data in Tables 2 and 3, it is preferable to apply a statistical test of comparison of the means to demonstrate that the means compared are statistically different or not in order to be able to adjust the comments on these statistical analysis results.
